Looking upstream: enhancers of child nutritional status in post-flood rural settings

Rodriguez-Llanes Jose Manuel 1 jmr.llanes@gmail.com jose.rodriguez@uclouvain.be
Ranjan-Dash Shishir 2 3
Mukhopadhyay Alok 4
Guha-Sapir Debarati 1
1 Centre for Research on the Epidemiology of Disasters, Institute of Health and Society, Université Catholique de Louvain , Brussels , Belgium
2 Department of Management, Siksha ‘O’ Anusandhan University , Bhubaneswar , India
3 Tata Trusts , Mumbai , India
4 Voluntary Health Association of India , New Delhi , India
Pérez-Jiménez Jara
Electronic publication date: 2016 Mar 1
Publication date: 2016
Volume: 4
Electronic Location ID: e1741
Received 2015 Dec 2; Accepted 2016 Feb 6
Copyright: ©2016 Rodriguez-Llanes et al.
Copyright year: 2016
Copyright holder: Rodriguez-Llanes et al.
License: This is an open access article distributed under the terms of the Creative Commons Attribution License, which permits unrestricted use, distribution, reproduction and adaptation in any medium and for any purpose provided that it is properly attributed. For attribution, the original author(s), title, publication source (PeerJ) and either DOI or URL of the article must be cited.
License URL: https://creativecommons.org/licenses/by/4.0/

Keywords: Flood, Climate change adaptation, Wasting, Stunting, Father education, Mother education, Parental education, Income, Maternal autonomy, Subsistence farming

Funding: European FP6 6th Framework Programme Social and Economic Impacts of Extreme Events: Evidence, Methods and Tools GOCE-CT-2007-036877 The present research was funded by the European FP6 6th Framework Programme under The MICRODIS Project—Integrated Health, Social and Economic Impacts of Extreme Events: Evidence, Methods and Tools (contract number GOCE-CT-2007-036877). The funders had no role in study design, data collection and analysis, decision to publish, or preparation of the manuscript.

==============================
Background. Child undernutrition and flooding are highly prevalent public health issues in many developing countries, yet we have little understanding of preventive strategies for effective coping in these circumstances. Education has been recently highlighted as key to reduce the societal impacts of extreme weather events under climate change, but there is a lack of studies assessing to what extent parental education may prevent post-flood child undernutrition.

Methods and Materials. One year after large floods in 2008, we conducted a two-stage cluster population-based survey of 6–59 months children inhabiting flooded and non-flooded communities of Jagatsinghpur district, Odisha (India), and collected anthropometric measurements on children along with child, parental and household level variables through face-to-face interviews. Using multivariate logistic regression models, we examined separately the effect of maternal and paternal education and other risk factors (mainly income, socio-demographic, and child and mother variables) on stunting and wasting in children from households inhabiting recurrently flooded communities (2006 and 2008; n = 299). As a comparison, separate analyses on children in non-flooded communities were carried out (n = 385). All analyses were adjusted by income as additional robustness check.

Results. Overall, fathers with at least completed middle education (up to 14 years of age and compulsory in India) had an advantage in protecting their children from child wasting and stunting. For child stunting, the clearest result was a 100–200% lower prevalence associated with at least paternal secondary schooling (compared to no schooling) in flooded-areas. Again, only in flooded communities, an increase in per capita annual household income of 1,000 rupees was associated to a 4.7–4.9% lower prevalence of child stunting. For child wasting in flooded areas, delayed motherhood was associated to better nutritional outcomes (3.4% lower prevalence per year). In flooded communities, households dedicated to activities other than agriculture, a 50–51% lower prevalence of child wasting was estimated, suggesting farmers and fishermen as the most vulnerable livelihoods under flooding. In flooded areas, lower rank castes were at higher odds of both child wasting and stunting.

Conclusions. In the short-term, protracted nutritional response in the aftermath of floods should be urgently implemented and target agricultural livelihoods and low-rank castes. Education promotion and schooling up to 14 years should have positive impacts on improving children nutritional health in the long run, especially under flooding. Policies effectively helping sustainable livelihood economic development and delayed motherhood are also recommended.

Introduction

Among all the disaster risks associated to a warming climate, flooding has become the most frequent and since the nineties it has been affecting grosso modo 100 million people a year. This is more than any other disaster type worldwide, climate-related or not (EM-DAT, 2015). Yet relative to their importance, the health consequences of flooding have been rarely investigated (Ahern et al., 2005; Alderman, Turner & Tong, 2012). Even if, at the time of writing, an equivalent 37.8% (i.e., 2.8 billion people) of the current world population were affected by floods in the past 25 years, little serious actions are being taken.

Education has been recently stressed as one key to reduce the societal impacts of climate change (Striessnig, Lutz & Patt, 2013). The study of educational attainment as a promotor of disaster resilience recently received substantive attention in a collection of 11 papers (Muttarak & Lutz, 2014). The results of this special issue are a convincing step forward. These studies were undertaken in different countries and regions, and targeted relevant outcomes in different phases of the disaster cycle in populations exposed to diverse climate-related hazards. Overall, they found that higher-educated groups avoided high-risk areas to settle, were better prepared, reacted more efficiently to early warnings and had lessen impacts on health and social variables, which indicated according to the authors better coping and recovery in these groups (Muttarak & Lutz, 2014). For the particular case of flooding, the evidence on the role of education as a promotor of positive health outcomes is scant and contentious (Lowe, Ebi & Forsberg, 2013). But what are the plausible pathways linking more education to health improvements? Indeed, formal education is crucial for acquisition and processing of information (e.g., literacy), improvement of cognitive abilities, decision making and long-term planning, and generally it leads to securing skilled jobs and ultimately higher income and better health (briefly reviewed in Striessnig, Lutz & Patt, 2013; Lindeboom, Llena-Nozal & Van der Klauuw, 2009).

The transgenerational effects of education on health have received substantial attention. Although the effect of maternal education on child stunting and general child health has been studied (Milman et al., 2005; Lindeboom, Llena-Nozal & Van der Klauuw, 2009), the effect of paternal education on child health has been more rarely examined (Moestue & Huttly, 2008; Semba et al., 2008). Assessing the extent to which the education of the father affects a child’s health is an important step to understand the relative contribution of fathers to the family wellbeing. Considering the father’s education is also a premise to further study the synergies between father and mother education levels in the promotion of child’s health (Semba et al., 2008).

Adequate child nutrition is a key indicator of wellbeing and development of particular importance in developing countries (Black et al., 2013). Nevertheless, studies connecting flood-exposure to the nutritional status of children have been rare, and none of them considered the role of father education in preventing the health impacts of floods among children (Phalkey et al., 2015). To the best of our knowledge, no study to date has investigated particularly the role of parental education on child’s nutritional status in post-flood settings. We only found one study which investigated the sole effect of the education of mothers, not the education of fathers, on malaria parasitemia in Sub-Saharan children (Siri, 2014). Importantly, no study has looked at the association of education on child wasting, let alone in the context of floods. The investigation of child wasting is relevant as evidence suggest that stunting and wasting represent different processes of undernutrition (Ricci & Becker, 1996). Recent evidence suggests that exposure to floods can be associated to increases in both child stunting (Rodriguez-Llanes et al., 2011) and child wasting (Rodriguez-Llanes et al., 2016) and thus both deserve attention in post-flood settings.

The large floods occurring in rural Odisha, India in September 2008 produced massive damage to agriculture, water and sanitation, communication networks and severe disruption to the normal functioning of the entire rural society at large (Government of Orissa, 2008). As part of an integrated project to investigate the health, social, and economic impacts of disasters, we carefully planned and conducted in September 2009 a representative survey of children affected and non-affected by these floods in 265 communities. To get further insight into prevention strategies for the health impacts of floods, which are absent in the literature (Bouzid, Hooper & Hunter, 2013) we examined in this study the effect of maternal and paternal education and other risk factors on stunting and wasting in children from families living in flooded and non-flooded communities.

Materials & Methods

Design and sample

Jagatsinghpur is a coastal district of the state of Odisha, located within the Bay of Bengal, India. The district’s population is around one million, 90% of which inhabit rural areas (The Census of India, 2011). The region is located in a large and fertile flood plain, crossed by large rivers, which makes it very attractive for fishing activities, livestock rearing and crop farming. However, this land is also regularly affected by heavy monsoons which often produce flooding. In the last decade, Jagatsinghpur has been hardly hit by five major floods, including coastal flooding associated to cyclone Paradip (05B) in 1999, followed by heavy rain floods in 2001, 2003, 2006 and the latest flood occurring prior our investigation, which took place in mid-September 2008 (Government of Orissa, 2008).

We conducted a population-based survey one year after the 2008 flooding in rural Jagatsinghpur district, Odisha, India. A two-stage cluster survey was necessary to obtain a probability sample of children 6–59 months of age in 265 villages from four severely flood-affected blocks of the district (Kujang, Biridi, Balikuda and Tirtol). Overall, 122 of these were flooded and 143 non-flooded in September 2008. The percentage of households flooded in each village was obtained for 2006 and 2008 events from OSDMA data (Odisha State Disaster Mitigation Authority, 2009; Fig. 1).

Figure 1 Study site, eligible villages and original sample of flooded and non-flooded villages in Jagatsinghpur district, Odisha, India.

Triangles represent flooded villages; circles those non-flooded. Size of polygons is proportionate to village size as measured by number of households (see map legend). Polygons overimpressed identified villages selected.

We used a two-stage cluster design to select our sample as our population of interest was clustered in villages and the information on the population was scant (Groves et al., 2009). In this study, the Primary Sampling Units (PSUs) were the villages and the children were the Secondary Sampling Unit (SSU). A 30 (PSU) by 30 (SSU) design, which should provide a probability sample of 900 children, was fixed. We initially enumerated the 265 villages along with each village population size projected for 2009 from census data (The Census of India, 2001), and subsequently 30 clusters from 29 villages were selected. This first selection was done using Probability Proportionate to Size (PPS) sampling with replacement (i.e., one large village was selected twice). This method picks up villages randomly, but the chances of selection are proportional to the size of the village, with selection probabilities favoring larger villages (Groves et al., 2009). At the second stage a fixed number of children (i.e., 30) are selected by cluster, independently of village size. The result is that chances are compensated with equal probabilities of selection of any eligible child listed in the sampling frame (Groves et al., 2009). Importantly, an updated list of eligible children was obtained from the ICDS centers (1 ICDS center for every 1,000 inhabitants) and validated with the ward members of each village. A total 3,671 eligible children were listed from 29 villages within a month prior training and piloting of the survey instrument. Once the lists compiled, it was detected that three flooded villages (Korana, Jamphar and Raghunathpur) and two non-flooded (Muthapada, Sureilo) did not have enough eligible children (i.e., less than 30). We created 5 new clusters by merging the list of children in each of these 5 villages (n = 280) with those of the closest non-selected flooded or non-flooded village, respectively, of our list (Fig. 1). Finally, thirty children per cluster were randomly selected.

Ethical approval

This study was approved by the Community Health Ethics Committee, Voluntary Health Association of India, New Delhi. Persons eligible to participate in the study were not offered any monetary incentive. Written informed consent was obtained for every head of household visited. In case the respondent was an illiterate, we asked a literate person from the community to read out the consent form and explain it to the head of the family. We then obtained the thumb impression of the respondent. In those cases, the person who read out the consent form also signed as a witness. Research procedures were consistent with the Declaration of Helsinki (1997). Interviews were administered after obtaining informed consent. The protocol was reviewed by a small group of scientists who had experience working with survivors of natural disasters and amended based on their recommendations.

Data collection: instruments and measures

Our survey instrument was adapted from the core one developed and approved by a multidisciplinary consortium of researchers from the MICRODIS project. The instrument development was based on interim literature reviews, and follows the UNICEF conceptual framework on child malnutrition (United Nations Children’s Fund, 1997). The questionnaire was validated prior to our study in research sites in India, Indonesia, the Philippines, Vietnam and the UK. We collected background information at the household level, more specifically from mothers and fathers, covering basic socio-demographic characteristics, wealth, child caring practices, healthcare access, maternal and paternal education, income and credit practices, water and sanitation, food consumption patterns; demographics, nutrition and health status data at the child level.

To assess nutritional status using anthropometric indicators, weight and height/length were recorded. Children were weighed without clothes. Weight was measured to the nearest 100 g by trained research assistants using a beam balance (<10 kg; Raman Surgical Co., Delhi-33, India) and an electronic balance for those children heavier than 10 kg. For children younger than 2 years of age, length measurements were taken to the nearest millimeter in recumbent position with an infantometer (Narang Medical, Delhi-110 028, India). If children were older than 2 years, they were measured standing up with an adjustable board calibrated in millimeters. Each research assistant measured and weighted each child twice to minimize measurement errors and use the average value of both measurements to gain precision. These instruments were calibrated daily.

Study questionnaires were administered by twelve experienced research assistants (Rodriguez-Llanes et al., 2011) from the Voluntary Health Association of India (VHAI) who received specific training on anthropometry and interview procedures for this study in late August 2009. The questionnaire was piloted in 12 households (6 in flooded villages and 6 in non-flooded) and improved based on the inputs of the pilot exercise. The study questionnaire was translated to the local language in Odisha (Oriya) and subsequently back translated into English by different professional translators, and a researcher checked the level of agreement between both versions. Duration of interviews ranged from 45 to 60 min, and all field work was completed between 6 and 24 September 2009.

Study variables

As our aim was to estimate the association effect of formal maternal and paternal education on child undernutrition, we excluded variables in our dataset which may have mediated this effect (Schisterman, Cole & Platt, 2009). We supported our choice by examining these variables (e.g., caregiving practices, food security, and access to health care, water and sanitation) framed within the UNICEF framework for child malnutrition (United Nations Children’s Fund, 1997). Overall 51 variables were assessed in a previous study (Rodriguez-Llanes et al., 2016) but for the purpose of this study we only analyzed distal determinants of child undernutrition.

Two outcomes were used in our study: stunting (height-for-age) and wasting (weight-for-height). Stunting is an indicator of chronic malnutrition, whereas wasting often evaluates acute nutritional stress at individual and population levels. The new WHO standard was used to calculate the z scores for these indicators. Malnutrition was a binary variable indicating whether a children is malnourished, z score ≤ 2 (1) or not (0) at the time of the interview.

Figure 2 Flow diagram of sampling procedure, sample obtained and analyses undertaken.

Overall, 17 variables were examined as potential predictors. The two fundamental variables of this study were the level of formal education attained by both children’s parents, mother and father. Formal education was the only assessed in this study, as we excluded parents with technical training or professional studies (Fig. 2). We studied education using the same categories as recorded in the original questionnaire, which follow benchmarks in the Indian educational system: never attended school (0 years of schooling), completed elementary school (5 years of schooling), middle school (8 years), high school (12 years) and completed university studies (15 years or more). For mothers, age at marriage, age at first delivery and age at birth of the selected child were reported and analyzed in years; the later calculated by subtracting the age of each child (converted to years) to the mother’s age. The father’s age at birth of the selected child was obtained by same calculations. Child’s sex was a binary variable. The age of each child in months was obtained from birth certificates and vaccination cards. If these were not available, local calendars were used. The birthweight of each child was recorded in grams from birth certificates and for analytical purpose we expressed birthweight per 100 g. The count of children younger than 5 years living in a household were used in our analyses. The principal means of livelihood were dichotomized as agriculture (taken as reference), grouping households dedicated to fishing, livestock rearing and crop farming versus non-agricultural for any other reported activity. Two religions were present in the study area: Hinduism, taken as the reference group, and Islam. The caste of the household was based on the household head and was grouped as a scheduled caste, other backward class or general class (reference category). The general class is the higher caste status. The scheduled caste is the social group historically subject to the higher deprivation levels in the country. Owned land was originally collected in acres but was expressed in hectares and analyzed as a continuous variable. Annual household income was recalculated per capita using data on household size and expressed per 1,000 Indian rupees (INR) and modeled as a continuous predictor. A household owning any livestock, including chicken, were modelled against those households not owning any. Finally, we recorded the exact number of persons residing in each household (household size) but were dichotomized as more than four or otherwise to account for overcrowding as done by a previous study (Semba et al., 2008). The levels of occupation (employed, unemployed or working as housewife) of mother and father were examined to better comprehend the level of gender empowerment within the household.

Statistical methods

Two expert data managers entered the data. An external researcher (JMR) run exploratory data analysis to identify errors in data entry and other implausible values (Day, Fayers & Harvey, 1998). ENA for SMART software (version November 2008) was used to calculate nutritional indicators with the 2006 World Health Organization Standard (ENA, 2008). ENA software is built up with specific functions for detection of outliers and impossible values. These were used to identify further problematic values among the calculated nutritional indicators. Each potential error discovered was discussed and investigated to determine where they originated and subsequently corrected.

The main predictor in this study was mother and father education. The original sample size was determined based on requirements of a previous study (Rodriguez-Llanes et al., 2016). Regarding the main research question addressed by this report, overall, available sample sizes were sufficient for subgroup analyses to detect prevalence ratios of two or more with a 80% power (Sullivan, Dean & Soe, 2009).

We examined the relationship between parental education and child nutritional status separately in repeatedly flooded and non-flooded cohorts. A first consideration was not to model maternal and paternal education jointly, but in separate analyses (Fig. 2). Intuitively, individuals with similar education tend to get married more often and this was reflected by our data: variance inflation factor (VIF) did show these two variables being highly collinear. As such, eight models were fitted on this data (2 datasets × 2 outcomes × 2 predictors each). These models are summarized in Fig. 2.

Bivariate and multiple adjusted logistic regression models with a quasi-binomial distribution to control for overdispersion were fitted (Lumley, 2010). For multivariate models, we included only variables with p < 0.1 in bivariate associations with undernutrition. We first ran a full model with all variables having a p-value lower than 0.1, a bit more conservative that some authors recommend (Vittinghoff et al., 2005). On each full model, the VIF on each predictor was calculated and predictors with the highest VIF were sequentially eliminated until all remaining predictors had a VIF lower than four. Backward selection was then applied to the remaining variables. At each step the non-significant variable with the largest p-value was excluded to obtain a final model including only significant variables at alpha level of 5% (i.e., p-value < 0.05). As proposed in recent literature (Siri, 2014; Muttarak & Lutz, 2014), we examined the impact of household annual per capita income on the effects on paternal education to ensure that education and income are independent contributors of child nutrition outcomes. We did that in all eight models.

Results were provided as prevalence ratios, crude [PR] or adjusted [aPR] with 95% confidence intervals. Given the limits of our sample size, interactions were not examined. All tests were two-tailed with α = 0.05. All analyses were weighted. Weights were calculated as the inverse of the selection probabilities. Statistical analyses were conducted in R (version 3.0.2) (R Development Core Team, 2008) with the survey package (Lumley, 2010).

Results

Study sample

A sample representative of the children population 6–59 months of age was obtained in 265 villages of Jagatsinghpur district. Our analyses excluded children with missing observations on relevant variables and from villages inundated in September 2008 but not in 2006. Figure 2 provides further details on each stage of the sampling and exclusions before statistical analysis.

Table 1 presents descriptive information for the variables analyzed in recurrently flooded communities (n = 299) and those non-flooded (n = 385). The illiteracy of mothers was less prevalent in flooded areas compared to non-flooded. Whereas more mothers completed middle school in flooded communities, more completed university studies in non-flooded ones. The percentage of fathers completing high school was 10% higher in flooded communities compared to non-flooded, while those having university degrees was similar. In general the flooded population was more educated relative to the non-flooded, and fathers were more educated than mothers. Almost 75% of the mothers attained at least middle school amongst the flooded villages relative to 67% in the non-flooded. For a similar comparison amongst fathers, the percentages were higher, 89% and 81.2%, respectively.

Table 1 Prevalence of stunting, wasting and baseline characteristics in rural flooded and non-flooded communities of Odisha, India.

	Flooded (n = 299)	Non-flooded (n = 385)	
Variables	n	% or mean (SE)	n	% or mean (SE)	
Stunting	299	30.4 (0.03)	385	29.0 (0.03)	
Wasting	299	51.5 (0.03)	385	20.3 (0.03)	
Maternal education					
None	17	5.4 (0.01)	37	9.5 (0.02)	
Primary school	59	19.9 (0.03)	96	23.5 (0.02)	
Middle school	106	36.0 (0.03)	105	26.1 (0.02)	
High school	85	29.3 (0.03)	108	28.7 (0.03)	
College or more	32	9.3 (0.02)	39	12.2 (0.02)	
Paternal education					
None	6	2.5 (0.01)	15	4.1 (0.01)	
Primary school	29	8.5 (0.02)	55	14.2 (0.02)	
Middle school	87	25.4 (0.03)	116	29.6 (0.03)	
High school	116	41.0 (0.03)	123	31.0 (0.03)	
College or more	61	22.6 (0.03)	76	21.2 (0.03)	
Mean age at marriage of the mother, years	299	21.8 (0.18)	385	21.5 (0.17)	
Mean age of the mother at first delivery, years	299	23.7 (0.17)	385	23.4 (0.17)	
Mean age of the mother at birth of selected child, years	299	26.1 (0.24)	385	25.9 (0.21)	
Mean age of the father at birth of selected child, years	299	31.3 (0.27)	385	31.1 (0.27)	
Mean annual income per household capita, 1,000 rupees	299	7.5 (0.60)	385	7.3 (0.63)	
Means of livelihood linked to agriculture	299	41.8 (0.03)	385	35.8 (0.03)	
Mean land owned, hectares	299	0.6 (0.05)	385	0.4 (0.04)	
Any livestock owned	184	64.7 (0.03)	209	55.2 (0.03)	
Religion					
Hindu	248	89.9 (0.01)	357	92.7 (0.01)	
Muslim	51	10.1 (0.01)	28	7.3 (0.01)	
Caste					
General	83	33.2 (0.03)	68	20.5 (0.03)	
Other backward	100	32.4 (0.03)	183	41.9 (0.03)	
Scheduled caste	66	24.6 (0.03)	106	30.3 (0.03)	
No caste	50	9.8 (0.01)	28	7.3 (0.01)	
No. of individuals eating from same kitchen >4	240	78.7 (0.03)	287	76.5 (0.02)	
Mean no. of children under five eating from same kitchen	299	1.4 (0.04)	385	1.5 (0.05)	
Child female	140	45.0 (0.03)	175	44.4 (0.03)	
Mean birthweight, 100 g	299	27.7 (0.25)	385	27.4 (0.21)	
Mean child age, months	299	33.4 (1.08)	385	32.5 (0.97)	

Flooded households showed higher reliance on agriculture activities as per their higher percentages of land and livestock owned, which corresponded to higher proportions dedicated to agricultural activities (41.8% in flooded vs. 35.8% in non-flooded). Households of the highest cast were also more common in the sample of flooded villages whereas schedule and other backward castes were found more often in non-flooded communities. Unemployment rates among fathers were 2.1% in non-flooded compared to 2.3% in recurrently-flooded populations. As many as 97.7% of all interviewed mothers worked as housewives in both flooded and non-flooded communities.

Correlates of child stunting

In univariate analyses, paternal and maternal education played an important role in reducing child stunting (Fig. 3). Overall, the protective effect of education on child stunting were substantial for most educated groups, amongst flooded communities and in men. Compared to mothers never attending school, a 3-fold (200%) lower prevalence of stunting was observed in most educated ones (i.e., those completing university studies). But generally the effects observed on paternal education were larger and stronger compared to maternal education (Fig. 3).

Figure 3 Factors associated to child stunting in repeatedly flooded and non-flooded communities of rural coastal Odisha, India.

Blue dots show the prevalence ratios in repeatedly flooded communities. Red dots in non-flooded. Dotted lines show the relative difference in the univariate effects between flooded and unflooded communities. ∗∗∗p < 0.001; ∗∗p < 0.01; ∗p < 0.05, p < 0.1.

An additional hectare of land owned in flooded communities had a substantial protective effect, although non-significant (p = 0.07; detailed results in Table S1), but the effect was lower in non-flooded. A per capita household annual income improvement of 1,000 rupees was also associated with a decrease of about 6% in child stunting, but again, only in flooded households (Fig. 3, Table S1). Belonging to a backward caste was detrimental to child stunting but only in flooded communities, and especially deleterious in the most deprived caste in India, the scheduled caste, with more than a twofold associated increase in the prevalence of child stunting.

Multivariate analyses on child stunting

In adjusted analyses, the positive effect of maternal and paternal education on child nutrition weakened and remained statistically significant only for models on paternal education in flooded communities, and marginally significant in non-flooded (Table 2). In repeatedly-flooded communities, fathers with completed middle, high school or university studies were consistently associated with 2.5-, 2.1- and 2.7-fold lower prevalence of child stunting, respectively. In those non-flooded communities, completed middle or high school by fathers conferred lower protection to their children, with associated 1.9- (p = 0.055) and 1.8-fold (p = 0.085) lower prevalence of stunting but none reached statistical significance. Having completed college was comparably as protective in non-flooded areas as in flooded (2.8-fold reduction), and the effect statistically significant (p = 0.015).

Table 2 Multivariate logistic regression models for maternal and paternal education and further risk factors associated to stunting in flooded and non-flooded children populations in rural Odisha, India.

	Flooded (n = 299)	Non-flooded (n = 385)	
	aPR (95% CI)	p-value	aPR (95% CI)	p-value	
Models for maternal education	–	–	–	–	
Maternal education	–	–	–	–	
No schooling	1	–	1	–	
Primary school	0.895 (0.558, 1.434)	0.645	1.275 (0.657, 2.475)	0.473	
Middle school	0.572 (0.335, 0.978)	0.042	0.927 (0.461, 1.866)	0.832	
High school	0.839 (0.468, 1.504)	0.556	0.523 (0.242, 1.127)	0.099	
College or more	0.598 (0.228, 1.567)	0.296	0.717 (0.263, 1.954)	0.516	
Child age (months)	1.015 (1.003, 1.027)	0.014	1.015 (1.004, 1.026)	0.008	
Caste	–	–	–	–	
General	1	–	NA	NA	
Other backward	1.654 (0.864, 3.164)	0.130	NA	NA	
Scheduled caste	2.007 (1.068, 3.770)	0.031	NA	NA	
No caste	1.560 (0.772, 3.151)	0.216	NA	NA	
Per head annual income (per 1,000 rupees)	0.955 (0.913, 0.998)	0.043	NA	NA	
Models for paternal education					
Paternal education	–	–	–	–	
No schooling	1	–	1	–	
Primary school	0.779 (0.478, 1.268)	0.315	0.523 (0.260, 1.052)	0.070	
Middle school	0.404 (0.247, 0.661)	<0.001	0.530 (0.278, 1.012)	0.055	
High school	0.487 (0.300, 0.624)	0.004	0.563 (0.293, 1.081)	0.085	
College or more	0.372 (0.205, 0.791)	0.047	0.360 (0.159, 0.816)	0.015	
Child age (months)	1.013 (1.001, 1.024)	0.033	1.012 (1.001, 1.023)	0.035	
Child birthweight (per 100 g)	NA	NA	0.950 (0.905, 0.997)	0.037	
Per head annual income (per 1,000 rupees)	0.953 (0.911, 0.996)	0.032	NA	NA	
Notes.

NA, results not available if variable not retained in multivariate model.

Per capita annual income was a consistent effect in flooded studied areas and across models on maternal and paternal education with 4.7%–4.9% lower prevalence of stunting for each yearly increase in 1,000 rupees per capita at household level. Schedule caste remained a relevant factor in multivariate analysis modeling the effect of maternal education but only in flooded communities (Table 2). The two final models for non-flooded communities were also adjusted for the effect of income but no substantial change in the coefficients were noted.

Correlates of child wasting

Crude associations showed an important role of parental education to reduce child wasting, with a preponderance of paternal over maternal effects of formal education and the largest effects observed in non-flooded communities (Fig. 4). An association between means of livelihood (non-agriculture vs. agriculture) and child wasting favoring households not relying on agriculture activities was observed, with a 47% lower prevalence of child wasting compared to those dedicated to agricultural activities (Fig. 4, detailed results in Table S2).

Figure 4 Factors associated to child wasting in repeatedly flooded and non-flooded communities of rural coastal Odisha, India.

Blue dots show the prevalence ratios in repeatedly flooded communities. Red dots in non-flooded. Dotted lines show the relative difference in the univariate effects between flooded and unflooded communities. ∗∗p < 0.01; ∗p < 0.05, p < 0.1.

Multivariate analyses on child wasting

The results for paternal education were different in adjusted models compared to those showed by bivariate associations (Table 3, Table S2 and Fig. 4).

Table 3 Multivariate logistic regression models for maternal and paternal education and further risk factors associated to wasting in flooded and non-flooded children populations in rural Odisha, India.

	Flooded (n = 299)	Non-flooded (n = 385)	
	aPR (95% CI)	p-value	aPR (95% CI)	p-value	
Models for maternal education	–	–	–	–	
Maternal education	–	–	–	–	
No schooling	1	–	1	–	
Primary school	0.817 (0.531, 1.257)	0.359	0.819 (0.435, 1.542)	0.536	
Middle school	1.013 (0.730, 1.407)	0.939	0.338 (0.157, 0.728)	0.006	
High school	0.858 (0.598, 1.231)	0.406	0.360 (0.178, 0.726)	0.005	
College or more	0.975 (0.609, 1.563)	0.917	0.637 (0.247, 1.643)	0.351	
Mother age at birth of selected child (years)	0.967 (0.944, 0.989)	0.005	NA	NA	
Means of livelihood (non-agricultural vs agricultural)	0.660 (0.526, 0.828)	<0.001	NA	NA	
Child age (months)	NA	NA	0.975 (0.956, 0.994)	0.009	
Caste	–	–	–	–	
General	1	–	NA	NA	
Other backward	1.396 (1.016, 1.917)	0.040	NA	NA	
Scheduled caste	1.272 (0.887, 1.826)	0.191	NA	NA	
No caste	1.393 (0.913, 2.130)	0.125	NA	NA	
Number of individuals eating from same kitchen	–	–	–	–	
2, 4	NA	NA	1	–	
>4	NA	NA	1.870 (1.059, 3.301)	0.032	
Models for paternal education					
Paternal education	–	–	–	–	
No schooling	1	–	1	–	
Primary school	0.777 (0.567, 1.064)	0.117	0.348 (0.144, 0.840)	0.019	
Middle school	0.571 (0.432, 0.754)	<0.001	0.349 (0.155, 0.787)	0.012	
High school	0.699 (0.579, 0.844)	<0.001	0.366 (0.175, 0.763)	0.008	
College or more	0.521 (0.363, 0.750)	<0.001	0.490 (0.220, 1.090)	0.081	
Means of livelihood (non-agricultural vs agricultural)	0.667 (0.526, 0.846)	<0.001	NA	NA	
Child age (months)	NA	NA	0.977 (0.958, 0.997)	0.023	
Notes.

NA, results not available if variable not retained in multivariate model.

In flooded communities, the protective effects of different levels of paternal education were all significant (p < 0.001) except for that of primary education (p = 0.117). However, larger effects were observed in the non-flooded communities surveyed in our study starting from primary education (2.9-fold lower prevalence, p = 0.019), middle (2.9 times lower, p = 0.012) or high school (2.7-fold difference, p = 0.008) up to those with university degrees who were associated with two times lower prevalence of wasting among their children, though the association remained insignificant (p = 0.081). In flooded households dedicated to activities other than agriculture, a 50–51% lower prevalence of child wasting was estimated (Table 3).

In modeling the effect of maternal education in flooded communities, we found that the mother’s age at birth of the studied child was associated with an improved nutritional outcome of their children. Each additional year was associated with a 3.4% lower prevalence of wasting (p = 0.005; Table 3). In flooded communities only and for the model on maternal education, belonging to other backward castes showed to be deleterious on child wasting (Table 3). All four final models in this section were also adjusted for annual per head income but none of those inclusions impacted our results.

Discussion

This study identifies education as one key investment in reducing the health impacts of extreme events under climate change risks while promoting sustainable human development. The most striking finding is that paternal education, and not maternal education, appeared to be the strongest predictor of lower child stunting and wasting in these communities. In non-flood settings, large samples from Indonesia and Bangladesh studied the effect of parental education on child stunting, and comparable positive effects were found for maternal and paternal education (Semba et al., 2008). The effect of paternal education on child wasting and stunting was also found to be comparable to that found in mothers in India and Vietnam (Moestue & Huttly, 2008). In flood settings, however, these effects were not as clear as in available studies. Hossain & Kolsteren (2003) found no effect of mother education on child wasting recovery four months after large floods in Bangladesh. Also in Bangladesh, Del Ninno & Lundberg (2005) found no effect of maternal education on child growth between three waves of anthropometric data collected within 15 months after the floods. None of these studies considered the effect of paternal education.

In our specific setting, we hypothesize that the observed differences in effects favoring paternal education could be partly explained by the low economic independence of women in our sample: nearly 98% of mothers in both exposure groups were housewives and we assumed that they had no revenues. We acknowledge that this topic is complex and more studies on this specific interaction should be conducted on datasets having larger samples than ours. However, in India evidence exist pointing to low maternal economic and physical autonomy as a contributor to child stunting (Shroff et al., 2009; Imai et al., 2014).

A second observation is that the effect of paternal education on child stunting were larger and statistically significant in flooded communities while the same estimated effects in non-flooded communities were of smaller magnitude. It looks as if the already positive effect of education might be boosted in a post-flood situation. Several studies have shown that education might substantially improve coping after disasters, through avoiding income loss after a disaster (Garbero & Muttarak, 2013), diversification of economic activites (Van der Land & Hummel, 2013) or choosing sustainable mechanisms for coping prevail over short-term views (Wamsler, Brink & Rantala, 2012; Helgeson, Dietz & Hochrainer-Stigler, 2013). In a disaster situation, decisions on allocation of the constraint resources available can be crucial. More educated parents might be able to invest in coping strategies with longer term benefits, have more savings or simply work in professions which are less affected by floods. Our results on child wasting showing that non-agricultural livelihoods withstood better the shock caused by flooding reinforce this point. Similar results on a higher likelihood of post-disaster migration in low educated farmers are consistent with our views (Van der Land & Hummel, 2013). Crop farming, livestock rearing and fishing were the most affected activities according to reports from the Government of Odisha: around half the production of the 0.44 Million Kharif crops flooded were lost, more than 2.3 Million livestock flood-affected, and above 6,300 fishing boats with their nets and other equipment damaged (Government of Orissa, 2008). Our findings are consistent with this pattern of affectedness and reveal their mid-term consequences on child nutritional health. Young children of agricultural livelihoods were the most impacted regarding the prevalence of wasting one year after the floods, which was around 50% higher than in other livelihoods. Importantly, no effect was observed for similar analyses conducted in non-flooded communities, whether univariate or multivariate. This reinforces the hypothesis that crop destruction and overall food insecurity is a very likely pathway to child undernutrition among the flooded populations (Leaning & Guha-Sapir, 2013). There is much to do to in terms of mitigation strategies to reduce the initial impact of floods on the livelihoods of these vulnerable populations. At the same time, there is an urgent need to reform and expand the relief response, which was not commensurate to the magnitude and duration of the problem. The basic needs of the affected were only covered by the government on the first 15 days following the floods. We need to ensure that these livelihoods are satisfied on around a year after the floods and that we target most affected livelihoods. Crops take months to be harvested since they are planted. Animal stocks need time to recover fully; the same applies for repairing boats or buying new ones. Without more dedicated resources, there is the risk of perpetuating poverty cycles.

Another consistent result was the positive effect of income on child nutrition in flood-affected communities. For example, an increase in 5,000 rupees per capita in yearly income within households should be associated with 25% lower prevalence of child stunting. Sustainable livelihood economic development is then a plausible strategy to reduce the nutritional burden of floods according to our data. In contrast, and again in flood-affected livelihoods only, lower caste was associated with worst nutritional status for both wasting and stunting variables, suggesting that caste is still in association with lower opportunities. Our study findings show that these social determinants of different impacts get visible in an extreme situation such as a disaster but we were unable to offer an explanation on the mechanism explaining this pattern in the data. Qualitative research might plausibly be very useful to investigate the reasons further. Finally, an important result was that mothers giving birth later had a lower likelihood of having a wasted children. Our model suggest that by delaying five years the birth of a given child, the associated prevalence of child wasting in case of floods would be on average 17% lower. It is noteworthy to point that this result was independent from the effect of mother education, adjusted for in this model.

Taking all the above together, education promotion in rural areas, especially among women, would be extremely efficient. In our study area about 30% of the mothers did not completed middle school education, which is mandatory in India. There is also an education gap with men in our study area, for whom this percentage was less than 20%, a 10% difference with women. It is worth mentioning that looking across all models in our study, primary education (up to 11 years of age) was not a significant contributor to better child nutrition in seven out of eight models reported. However, providing three additional years of schooling (up to 14 years) had a significant positive impact on child nutrition. Moreover, the magnitude of effects reported for middle, high and college education were quite comparable, suggesting that at least in these rural communities, middle education is sufficient. Our results are in agreement with the Indian policy on education. More efforts are then needed to ensure that every Indian children receives the eight years of compulsory formal education. By promoting more education and adequate family planning (Van Braeckel et al., 2012), women might naturally delay motherhood, with potential benefits to children’s nutritional health. To ensure full benefits, we hypothesized that mother education might lead to increased employment rates amongst women, as this would also contribute to additional higher household income (with associated benefits shown by our study). Notably, it has been demonstrated that income allocation by mothers is more efficient to improve the family’s health, including child nutritional status, compared to fathers (Thomas, 1990).

Strengths and limitations of the study

There a few methodological considerations, strengths and limitations that need to be addressed here. Firstly, all the reported results were adjusted, as an additional robustness check, by annual per capita household income. As such, all effects reported were independent of income, and thus our results can be methodologically comparable to those recently published (Muttarak & Lutz, 2014). Second, we often found in the literature maternal and paternal education jointly modelled (Semba et al., 2008). However, we found that these two variables were highly correlated in our data and enough evidence using variance inflation factor analysis, which supported that they should be modelled separately. Future studies should be aware of this. Third, father education was highly correlated to caste and mother’s age at birth of a child, and this is the reason why models modeling father education do not include these variables. Strengths of this study include carefully consideration of income, as well as having analyzed data on a non-flooded group which allowed us to observe that some effects such as a mother’s age at birth, income or caste become important determinants only in the extreme circumstances of flooding. Additional positive features of this study were its population-based design, the consideration of recurrently flooded communities only, coupled with the analysis of mid-term and more long-term child undernutrition. On the minus side, our study was not based on a very large sample size, which could have been missed the detection of smaller effects and provided tighter confidence intervals. Similarly, a larger sample size might have allowed us to analyze most severe forms of malnutrition, such as severe wasting and stunting. In the aftermath of flooding in Bangladesh, Choudhury & Bhuiya (1993) have shown maternal education to play a role in increasing severe forms of underweight. Also, as the findings presented are based on cross-sectional data, care is needed to not attribute causal relationships to the associations showcased here. Importantly, it is difficult to generalize our results to other settings such as urban slums, as vulnerabilities might be different there. However, we are confident that there might be similarities between our results and other rural areas around the world in which subsistence farming is the norm. With its limitations, this is one of the first studies to look at child undernutrition in flood settings and study potential preventive factors in short and long-term. More studies are needed to solidify the evidence base for global action.

Conclusions

We recommend strengthening the response to floods and targeting most vulnerable agricultural-dependent livelihoods, which showed an associated 50% higher prevalence of child wasting. Policies for relief should be reviewed to ensure a longer period of support and proper take up by development programs, which should give enough time and support to these households relying on agricultural activities to fully recover.

Education promotion in general and the strict fulfillment of schooling until the compulsory 14 years of age is strongly recommended to protect child’s health in the face of future flooding. Policies effectively helping sustainable livelihood economic development and diversification of economic activities, delayed motherhood, which certainly include education promotion, are vividly praised.

Given that global climate changes are set to increase flooding both in frequency and severity which will further aggravate this situation, we recommend urgent action to be taken (UNICEF Office of Research, 2014).

Supplemental Information

Supplemental Information 1 Supplemental Information

Click here for additional data file.

Data S1 Dataset used in the study

It includes 17 covariates, 2 outcomes, and flood exposure level, village and block names, and sampling probabilities required in the weighted analysis. Description of the data is included. Transformation of the variables are explained in the manuscript.

Click here for additional data file.

We are grateful to Jean Macq, Niko Speybroeck, Sophie Vanwambeke and Philippe Donnen for comments and guidance. We are grateful to Pascaline Wallemacq for preparing the map.

Additional Information and Declarations

Competing Interests

Author Contributions

Human Ethics

Data Availability

The authors declare there are no competing interests.

Jose Manuel Rodriguez-Llanes conceived and designed the experiments, analyzed the data, contributed reagents/materials/analysis tools, wrote the paper, prepared figures and/or tables, reviewed drafts of the paper, had the research idea, interpreted results, contributed data to map.

Shishir Ranjan-Dash conceived and designed the experiments, performed the experiments, contributed reagents/materials/analysis tools, reviewed drafts of the paper, contributed data to map.

Alok Mukhopadhyay conceived and designed the experiments, performed the experiments, reviewed drafts of the paper.

Debarati Guha-Sapir conceived and designed the experiments, reviewed drafts of the paper, gave advice, contributed to interpreting results.

The following information was supplied relating to ethical approvals (i.e., approving body and any reference numbers):

This study was approved by the Community Health Ethics Committee, Voluntary Health Association of India, New Delhi. An approval letter was issued by the Ethics Committee.

The following information was supplied regarding data availability:

The raw data has been supplied as Data S1.

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
