# Peer review of "Looking upstream: enhancers of child nutritional status in post-flood rural settings"

_PeerJ, doi:10.7717/peerj.1741_

## Round 0.1 · original submission · Major Revisions

· Academic Editor

Major Revisions

I think this manuscript deals with an interesting topic that fits within the scope of PeerJ. However, I consider the authors should carefully read the reviewers' comments, specially from Reviewer 1, since substantial modifications should be made in order to make it acceptable for publication.

Reviewer 1 ·

Basic reporting

1. Revise the PeerJ policies. In Ethical approval (lines 148-157), please include the approval reference numbers. If an approval reference number is not provided, written approval must be provided as a confidential supplemental information file according with the Journal policies and procedures. As consent was written, an empty copy of the consent form used should also be provided as a Confidential Supplemental Information file.
2. The style and English should be revised, sometimes the text is confusing, examples are provided in coming paragraphs.
3. The document include too many figures and tables. Figure 1 can be dropped, this figure does not provide practical/substantial information.
4. The introduction need to be revised and corrected: Line 58, drop in the past and remain today. Consider avoid using too. Line 66, What the authors refer to when they write In this study? Please rewrite. The problem is not well defined, the introduction is not clear and is difficult to read. The ideas are suddenly broken and later on in the text some ideas tend to be addressed again. The authors should clearly describe the problem. The importance of parental education on child malnutrition should be presented in a strong and consistent way. The authors suggest that only few studies have studied the separate associations of mother and father education with children undernutrition. But it is not clear from the introduction why we should expect different associations. Similarly why to expect different association for stunting/wasting. What is the rationale behind these hypothesis? Line 76, a new paragraph is written, but it seems that the ideas presented are still related with the previous paragraph, Consider to merge the two paragraphs and summarize the information presented. Lines 196-202. Background data should not be presented in the methods section, integrate this paragraph in the introduction. Line 76. Indeed seems not to be correct as it is preceded by a question….Line 84 revise the writing
5. The structure is in concordance with the Journal format. Still some recommendations per section:
ABSTRACT
Background, although parental education was an important predictor in this study, the importance of parental level of education is not addressed.
Methods and materials: in line 29, please specify the exact year as not everyone is familiarized with the year of the last large floods. In the same line, please write population-based survey.
Line 30, the figure 17.000 is irrelevant and confusing, better to include the sample size included in the analysis.
Line 33, drop “in this study”
The study design is not specified
Results: Some outcomes are presented in the results but are not in the methods

Include a description of the education system in Jagatsinghpur as it differs from one country to another. Include the number of years at each level in the description of the setting. Also explain here that middle education is mandatory in India.

Experimental design

6. The authors present an original research question within the scope of the Journal, still the question should be better justified in the introduction section as described in comment number 4. Although the authors present a research gap, the rationale behind the gap is not clear (see comment number 4).
7. Some methodological issues should be clarified
Line 102. Design and sample.
In the STROBE statement check list, the authors indicate that the study size calculations are not presented as the paper is based on existing data. This is not clear in the methods. Still, the authors should explain the sample size calculations and clarify if the sample size is sufficient for the new research question. Is the sample size sufficient to analyze the data separately for flooded / non-flooded communities?
The authors explain that a 30 by 30 design was used with the villages as the PSU. In line 122 it is written: and subsequently 30 clusters (29 villages). It is not clear to me why only 29 villages are included. It seems form the flow chart that 30 clusters were selected from 29 villages. But if the villages were the PSU it is not clear how the selection was done and what the clusters are

Validity of the findings

Line 158 Data collection and instruments. Was the questionnaire validated? Where the anthropometric measurements standardized? The authors do not explain if the children wore clothes during the measurements. The authors explain that measurements were taken twice, by the same researcher?? Or by different researchers? Line 181, how was agreement evaluated?
Study variables. Line 191, explain that malnutrition was evaluated by means of two outcomes: stunting and wasting. Explain which variables are predictors/cofounders
Statistical methods Line 228. How the data was entered and managed? How the authors ensured accuracy during data entry? Line 231, Which relationship?

Results. As the authors explain 900 children should be included. Only 684 children are included in the analysis 299 from flooded villages and 385 from non-flooded villages. Is this sample size sufficient? In The STROBE statement check list the authors explain that missing data were rare and that it is explained how missing data were addressed. I do not find this explanation. Include N at the beginning of the results.
In the flowchart the authors explain that 105 records from 5 villages not included in the original PPS were excluded. How is it possible that measurements were taken in these villages??? Is it related with the 5 villages not having enough eligible children?
It would be interesting to see the prevalence of stunting and wasting in flooded, non-flooded communities in Table 1
Explain the content of the Supplementary tables in the text
Line 294 the results are not substantial, they are not significant
Line 305, Do the authors refer to parents? Explain
Line 310 replace 2.7 by 2.8
In Table 2 and 3, explain why some data is not shown (NA)
Evaluate the need to present separate results for stunting/wasting, this makes the paper too long and difficult to follow. A strong rationale for a separate analysis should be included in the introduction section to maintain the analysis as currently presented.
DISCUSION
Line 387 results should not be presented in the discussion section

Reviewer 2 ·

Basic reporting

The submission adhiere to the PeerJ policies, is well written and include sufficient introduction and background. Relevant prior literature is appropriately referenced. The structure of the submitted article is conform to the template. Figures are relevant to the content of the article, with sufficient resolution and appropriately described.
The submission represent an appropriate unit of publication.
Lines 242-248 (pages 12 and 13), should correct the color of the font.
Lines 295 and 297 (page 16), "Table S1", should be changed to "Table 1".

Experimental design

The submission describe an original primary research within the Scope of the journal. It is clearly defined the research question, which is relevant and meaningful in the public health.

Validity of the findings

No Comments

Additional comments

The work presented is important because they are few studies that relate climate change to flooding, the consequences on the health of populations, specifically the nutritional status, and the lifestyle of families. Data analysis is robust and reliable. This is one of the first studies to look at child undernutrition in flood settings and study potential preventive factors in short and long-term.

Annotated reviews are not available for download in order to protect the identity of reviewers who chose to remain anonymous.

---

## Round 0.2 · accepted · Accept

· Academic Editor

Accept

I think the authors have properly answered the reviewers's comments and included in the mansucript the changes needed. Therefore, I think it is now suitable for publication.